# Atmospheric cold pools abruptly reverse thermohaline features in the ocean skin layer

5 Lisa Gassen<sup>1</sup>, Samuel M. Ayim<sup>1</sup>, Leonie Jaeger<sup>1</sup>, Jens Meyerjürgens<sup>1</sup>, Mariana Ribas-Ribas<sup>1</sup>, Oliver Wurl<sup>1</sup>

<sup>1</sup>Center of Marine Sensors, Institute for Chemistry and Biology of the Marine Environment, Carl von Ossietzky University of Oldenburg, Wilhelmshaven, Germany

Correspondence to: Lisa Gassen (<u>lisa.gassen-bertzbach@uol.de</u>)

**Abstract.** The ocean skin layer, which covers the upper millimetre of the sea surface, regulates the exchange of heat, gases, and freshwater between the atmosphere and the ocean. However, there is a lack of small-scale mechanistic understanding of these fluxes, especially under abrupt meteorological shifts, due to observational challenges during stormy conditions in the open sea. This study provides unique data on temperature and salinity anomalies between the skin layer and a depth of 100 cm during atmospheric cold pools, which induce abrupt shifts in air temperature, wind speed, precipitation, and heat fluxes. We determined how these abrupt meteorological shifts forced the anomalies and altered the conditions at the air-sea boundary layer during three events monitored by an autonomous surface vehicle. Two cold pool events were observed in the harbour of Bremerhaven and one event in the North Sea. Here, we show that the skin layer instantly reacts to abrupt meteorological shifts forced by cold pools. The average temperature change in the skin layer was twice as much as at a depth of 100 cm. An abrupt change in meteorological conditions, shifting the net heat flux from positive to negative, can turn a warm skin layer into a cooler layer compared with the 100 cm depth. Salinity anomalies in the harbour were less affected by abrupt meteorological shifts, including freshwater fluxes, than those in the North Sea event. The current velocities showed that changes in wind direction could alter the surface current direction, and that the backscatter signal consistently reflects wind-induced mixing, with higher backscatter observed during increased wind conditions. This study reveals the complex relationships between atmospheric conditions and oceanic responses and provides valuable information for understanding air-sea interactions and their implications for climate dynamics.

## 1 Introduction

The skin layer of the ocean represents the boundary between the atmosphere and the ocean and typically has a thickness of approximately 1 mm (Liss et al., 1997; Wurl et al., 2009). The skin layer is responsible for regulating the exchange of gases, heat, and freshwater between the atmosphere and the ocean (Fairall et al., 1996c; Liss and Duce, 1997; Gassen et al., 2023). It is usually cooler than the underlying near-surface layer (NSL) (first upper 100 cm) because of heat loss from the ocean to the atmosphere (Katsaros and Buettner, 1969; Schluessel et al., 1990; Zappa et al., 1998), making the skin temperature a critical indicator of gas and heat flux exchange processes (Fairall et al., 1996c). This cool skin layer remains in the presence of a diurnal thermocline (Minnett, 2003; Jessup et al., 1997; Yu, 2007). The cooling effect significantly affects gas solubility in the skin layer (Watson et al., 2020). A warmer skin develops in the case of a positive net heat flux within the skin layer (Katsaros, 1980).

Skin salinity is a key parameter for monitoring freshwater fluxes (evaporation–precipitation) over the ocean; adding or removing freshwater to or from the surface through precipitation and evaporation results in changes in surface salinity (Durack, 2015; Gassen et al., 2023; Gassen et al., 2024b). In the same processes, freshwater fluxes affect skin temperature. Evaporation releases heat from the sea surface into the atmosphere. Conversely, precipitation typically has a different temperature from the ocean surface, potentially altering the surface temperature (Gosnell et al., 1995; Williams and Stanfill, 2002). Precipitation typically enhances the gas exchange between the ocean and atmosphere (Parc et al., 2024; Ho et al., 2004), and the global effect is described as an increase in the oceanic sink for CO<sub>2</sub> of 6 % (Ashton et al., 2016).

Considerable research has been conducted on thermal stratification induced by diurnal warming, heat exchange processes, and wind-driven turbulent mixing in the NSL (Price et al., 1986; Fairall et al., 1996c; Gentemann et al., 2003). However, there have been comparatively few investigations into the effects of these processes on the skin layer. To achieve reliable outputs for atmospheric models of ocean heat fluxes, an accuracy of approximately 10 W m<sup>-2</sup> is required (Fairall et al., 1996a). The scarcity of in situ measurements on the skin layer is a significant challenge, particularly given difficulties in observing them during extreme meteorological events. Research vessels typically avoid deploying oceanographic equipment during unfavourable weather conditions.

Atmospheric cold pools, defined by air masses cooled often due to precipitation, cause sudden meteorological shifts and strongly influence upper ocean processes. These cold pools form when rain-cooled air descends, forming a dense air mass that can spread across vast oceanic areas (Zuidema et al., 2017). Cold pools drive mesoscale organization in cloud patterns by lifting moist air at their leading edges, forming arch-like cloud structures that allow clouds to develop deeper and broader, fostering convection in otherwise stable environments (Schlemmer and Hohenegger, 2014; Zuidema et al., 2017). Cold pools also impact the marine boundary layer by altering surface fluxes. For instance, increased wind and surface fluxes, including latent and sensible heat, are observed within cold pool fronts, contributing to temperature and velocity fluctuations that enhance surface energy exchange (De Szoeke et al., 2017). Moreover, the interaction of cold pools with the ocean surface can affect

the upper ocean's thermal and moisture dynamics, helping to regulate the exchange of heat and moisture between the ocean and atmosphere, thus playing a crucial role in the broader climatic system (De Szoeke et al., 2017; Zuidema et al., 2017). Measurements of skin layer temperature and salinity during rapid meteorological shifts require high temporal and spatial resolutions to understand immediate response mechanisms. Global climate change is increasing the frequency of extreme meteorological events (Allen and Ingram, 2002; Ummenhofer and Meehl, 2017), making observations during abrupt shifts critical for estimating heat flux changes. The consequences of heat, gas, and freshwater fluxes during rapid meteorological changes remain uncertain. Abrupt meteorological changes are often associated with increasing wind speeds (Webster et al., 1996; Ten Doeschate et al., 2019). While wind speed effects on thermohaline stratification have been studied (Leibovich, 1983; Soloviev et al., 2014), little is known about skin layer dynamics and fast mixing processes. Studies show the skin layer is affected by freshwater fluxes (Wurl et al., 2019; Gassen et al., 2023; Gassen et al., 2024a), and surface stratification is disrupted at wind speeds above 5 m s<sup>-1</sup> (Moulin et al., 2021; Gassen et al., 2024b). However, the effects of abrupt meteorological changes on temperature and salinity anomalies between the skin layer and 100 cm remain poorly characterized. Detailed observational data on rapid meteorological effects on skin layer dynamics are needed to improve climate model accuracy (Renfrew et al., 2002).

This study aims to investigate the effects of rapid meteorological changes, i.e. cold pools, on temperature and salinity anomalies, addressing a gap in the current understanding of skin layer dynamics under extreme meteorological conditions. Our measurements include recording relevant meteorological variables, such as wind speed and air temperature, and calculating heat and freshwater fluxes. Furthermore, our study measured near-surface currents and observed temperature and salinity in the skin layer and the NSL. These measurements were conducted during rapid meteorological changes in the harbour of Bremerhaven, Germany, and in the North Sea. Even though *in situ* observation data during rapid meteorological changes were extremely rare, we were able to record three events. The responses of temperature and salinity anomalies to rapid meteorological changes were determined. The consequences of heat and freshwater fluxes in the skin layer and mixing processes were also analysed. This approach aims to contribute to a more nuanced understanding of skin layer dynamics and to enhance the accuracy of atmospheric models by providing empirical data on the skin layer's responses to sudden meteorological changes. Our findings can improve and refine future observational strategies for air—sea interactions and models.

## 2 Materials and Methods

During the cruise HE614 of RV *Heincke*, temperature and salinity changes were recorded using the autonomous surface vehicle (ASV) *HALOBATES* (Wurl et al., 2024) during three events of rapid meteorological changes. Two events occurred in the harbour of Bremerhaven (53.52°N, 8.58°E) on 14 and 15 March 2023, respectively. The ASV can be operated manually with a remote control or autonomously through a pre-programmed mission. Due to space constraints in the harbour, the ASV was operated manually using a remote control within a few hundred metres of the berth of the research vessel (Fig. S1). An abrupt

meteorological change event during a previous RV *Heincke* cruise (HE609) in the North Sea (53.85°N, 7.68°E) was also analyzed. On 10 October 2022, the ASV was operated when a rapid meteorological change occurred. The ASV drifted during operational time (Gassen et al., 2024b). The data on temperature and salinity anomalies forced by precipitation for this event are presented and discussed in Gassen et al. (2024b). This study examines these data in the context of heat fluxes influenced by the cold pool.







Here, we referred to abrupt meteorological shifts as changes in wind speed of  $> 0.3 \text{ m s}^{-1} \text{ min}^{-1}$ . The ASV was equipped with six rotating glass discs. The skin of the water surface adhered to the glass discs (Shinki et al., 2012; Ribas-Ribas et al., 2017) and was sampled in a flow-through system to continuously measure temperature and conductivity (Ocean Seven310, Idronaut, Italy), which were later used for absolute salinity calculations. The sensors have an accuracy of 0.0015 °C and 0.0015 mS cm<sup>-1</sup> for temperature and conductivity, respectively. Water from a 100 cm depth was sampled and pumped through the temperature and conductivity probes (Wurl et al., 2024). Temperature and salinity anomalies ( $\Delta T$ ,  $\Delta S$ ) were calculated as the difference between the values from the skin layer ( $T_{skin}$ ,  $S_{skin}$ ) and those from the 100 cm depth ( $T_{100 \text{ cm}}$ ,  $S_{100 \text{ cm}}$ ):

$$\delta \quad \Delta T = T_{skin} - T_{100 cm} \tag{1}$$

$$\Delta S = S_{skin} - S_{100 cm} \tag{2}$$

Density was calculated using the Thermodynamic Equation of Seawater 2010 (TEOS-10) and the Gibbs-Sea Water Oceanographic package for R (Mcdougall and Barker, 2011). *HALOBATES* has two weather stations at a height of 2 m to determine meteorological variables, such as air temperature, humidity, and wind speed. One of the weather stations (Campbell Scientific Ltd., Model CR1000X, United Kingdom) measures at 10 s intervals, and the other (Thies Clima, Model SMP6, Germany) measures at 10 min intervals. For the station in the North Sea, only data from the Thies Clima weather station were used because of a technical failure of the weather station from Campbell Scientific.

Two pyrgeometers and pyranometers (IR20 (spectral range: 4.5 μm - 42 μm) and SR20-D2 (spectral range: 0.28 μm - 3 μm); Hukseflux, Netherlands) were installed on the RV *Heincke* to measure the downward and upward longwave and shortwave irradiances. To compute the heat fluxes and evaporation rates, the COARE 3.6 algorithm was applied (Fairall et al., 1996b; Fairall et al., 2003; Edson et al., 2013). Net heat flux was calculated as the sum of the longwave, shortwave, latent and sensible heat fluxes. Given that the measuring interval of the radiometers was set to 10 min, linear interpolation was applied to obtain values at 10 s intervals for the harbour events. An optical laser disdrometer (Thies Clima, 5.4110.xx.x00, Germany) was installed on the upper deck of the research vessel to monitor precipitation intensity and accumulated precipitation. Acoustic Doppler current profiler (ADCP) data were recorded at a frequency of 2 Hz, providing accurate estimates of current velocities that were referenced to the GPS position data. A 600-kHz Teledyne RD Instruments RiverRay ADCP was mounted at the centre and beneath the ASV and was configured with a 0.2 m bin size and a blanking distance of one cell (0.2 m) beyond the immersion depth of 0.3 m. This configuration provided the first data bin at a depth of 0.5 m.

A Pearson correlation analysis was conducted with the derived temperature and salinity anomalies and essential variables, such as air temperature, wind speed, precipitation intensity, net heat flux, and backscatter velocity. The results were considered

significant when p < 0.05, with a 95% confidence level. The correlation was considered strong, with correlation coefficients ranging from 0.7 to 1 or from -0.7 to -1. The duration and intensity of the meteorological changes differed between the harbour and North Sea events. For this reason, a correlation analysis of all the main variables with possible relationships with temperature and salinity anomalies was conducted separately for each event.

## 130 **3 Results**



To investigate the effects of rapid meteorological shifts caused by cold pools on temperature and salinity anomalies in the skin layer and at 100 cm depth, we first present the thermohaline features, including air and water temperature, salinity, density, wind speed, heat fluxes, precipitation, and evaporation, during two events in the harbour of Bremerhaven and one in the open North Sea. In one harbour event (15 March 2023), three sudden changes in wind speed were observed (A, B, and C in Table 1). Table 1 also summarizes the total changes in temperature and salinity, as well as the variations in meteorological conditions during all abrupt shifts. We then examine near-surface current velocities during these events and, finally, analyse the correlations between temperature and salinity anomalies and the meteorological variables.

Table 1: Total change in temperature and salinity of the skin layer and the 100 cm depth, and the change in meteorological variables during all abrupt weather shifts.

| Event      | Change in Skin | Change in 100 | Change in      | Change in 100   | Change in air | Change in    | Change in                | Change in           |
|------------|----------------|---------------|----------------|-----------------|---------------|--------------|--------------------------|---------------------|
|            | temperature    | cm            | Skin salinity  | cm salinity     | temperature   | wind speed   | precipitation            | net heat            |
|            | (°C)           | temperature   | $(g\ kg^{-1})$ | $(g \ kg^{-1})$ | (°C)          | $(m s^{-1})$ | intensity (mm $h^{-1}$ ) | $flux\ (W\ m^{-2})$ |
|            |                | (° <b>C</b> ) |                |                 |               |              |                          |                     |
| 14 March   | -0.57          | -0.37         | +0.10          | +0.12           | -4.3          | +8.55        | +7.26                    | -121                |
| 2023       |                |               |                |                 |               |              |                          |                     |
| 15 March   | -0.32          | -0.74         | -0.16          | -0.43           | -0.5          | +6.53        | +0.09                    | -81                 |
| 2023 - A   |                |               |                |                 |               |              |                          |                     |
| 15 March   | -0.29          | -0.08         | -0.15          | -0.09           | -1.4          | +10.69       | +0.37                    | -182                |
| 2023 - B   |                |               |                |                 |               |              |                          |                     |
| 15 March   | -0.40          | -0.23         | +0.10          | +0.06           | -2.4          | +9.21        | +1.24                    | -252                |
| 2023 - C   |                |               |                |                 |               |              |                          |                     |
| 10 October | -0.16          | -0.07         | -0.17          | -0.03           | -1.7          | +9.20        | +10.37                   | -100                |
| 2022       |                |               |                |                 |               |              |                          |                     |

## 3.1 Harbour event on 14 March 2023





## 3.1.1 Thermohaline features during abrupt meteorological changes

Meteorological conditions changed abruptly during the first event on 14 March 2023 in the harbour of Bremerhaven between 08:18 and 08:31 UTC (Fig. 1e–g). This was caused by rapid shifts in meteorological variables, such as wind speed and air temperature (Fig. 1e). Before the arrival of the weather front at the location of the ASV, the average air temperature remained constant at 9.24 ± 0.06 °C. With the approach of the cold pool, there was a notable decrease in air temperature between 08:19 and 08:25 UTC (i.e., within 8 min), reaching a minimum of 4.30 °C. Afterwards, the air temperature decreased further, but slower, from 4.9 °C to 2.9 °C within 40 min (Fig. 1e). Before the abrupt weather change, light precipitation and wind speed were observed, with mean intensities of 0.26 ± 0.24 mm h<sup>-1</sup> and 4.41 ± 1.19 m s<sup>-1</sup>, respectively (Fig. 1e, g).

During the abrupt meteorological change, the precipitation rate and wind speed increased notably by 7.26 mm h<sup>-1</sup> and 8.55 m s<sup>-1</sup>, respectively. Simultaneously, there was a change in wind direction from South–South–West to North–North–West (Fig. S2). Following the abrupt meteorological change, the mean wind speed remained high at  $7.40 \pm 1.94$  m s<sup>-1</sup>, with a maximum wind speed of 12.24 m s<sup>-1</sup> at 08:39 UTC. In general, the precipitation rate was relatively low, with a mean intensity of  $0.39 \pm 0.36$  mm h<sup>-1</sup> after the abrupt meteorological change (Fig. 1g). At 08:21 UTC, the evaporation rate shifted from negative to positive values—that is, from condensation to evaporation. Prior to the abrupt meteorological change, the mean evaporation rate was  $-0.01 \pm 0.01$  mm h<sup>-1</sup>, and it increased to  $0.05 \pm 0.01$  mm h<sup>-1</sup> after the meteorological change (Fig. 1g). The accumulated precipitation exceeded the accumulated evaporation during the operational time of the ASV (Fig. S3).

During the abrupt meteorological shift on 14 March 2023, the sea surface temperature decreased, whereas the salinity and density increased following an abrupt change in weather conditions (Fig. 1a–c). Before the abrupt meteorological shift (Fig. 1e–g), the skin layer was cooler than at the 100 cm depth, with a mean temperature anomaly of  $-0.09 \pm 0.03$  °C (Fig. 1d). From 08:19 to 08:24 UTC, a rapid decrease in temperature was observed across the skin layer and 100 cm depth, with the most notable decrease occurring in the skin layer (-0.50 °C). Following the abrupt meteorological change, cooler skin developed, exhibiting a maximum skin temperature anomaly of -0.44 °C. After the abrupt meteorological change at 8:18 UTC, the mean skin temperature anomaly during the subsequent period (08:28–09:29 UTC) was  $-0.35 \pm 0.04$  °C. At 09:15 UTC, the skin temperature increased, which caused an increase of approximately 0.1 °C in the temperature anomaly.

Before the abrupt meteorological shift, the salinity exhibited minimal variation across the skin layer and the 100 cm depth at a mean skin salinity of  $9.34 \pm 0.04$  g kg<sup>-1</sup>. After the abrupt change in meteorological conditions (Fig. 1e–g), a notable increase in salinity was observed at all depths at 08:25 UTC. At 08:28 UTC, there was a decrease in salinity for 2 min, followed by a subsequent increase. Following the abrupt meteorological change, the mean absolute skin salinity reached a value of  $9.47 \pm 0.44$  g kg<sup>-1</sup>. Throughout this occurrence, the salinity of the skin was consistently higher than that at the 100 cm depth. Consequently, the salinity anomaly remained positive—that is, the salinity of the skin remained continuously saltier than that at the 100 cm depth (Fig. 1d). The density graphs mirrored the pattern observed in absolute salinity, indicating an increase in density in the skin layer and at the 100 cm depth following an abrupt meteorological change (Fig. 1c). The mean density was

 $1007.41 \pm 0.25 \text{ kg m}^{-3}$  in the skin layer and  $1007.36 \pm 0.25 \text{ kg m}^{-3}$  at the 100 cm depth for the entire operation time, respectively.

Before the abrupt meteorological change, all surface heat fluxes were positive, thus warming the ocean (Fig. 1f), with a mean latent heat flux of  $10 \pm 4$  W m<sup>-2</sup>, sensible heat flux of  $22 \pm 8$  W m<sup>-2</sup>, longwave heat flux of  $3 \pm 1$  W m<sup>-2</sup> and overall net heat flux of  $79 \pm 14$  W m<sup>-2</sup>. With the onset of the cold weather front, the heat fluxes turned, resulting in a mean latent heat flux of  $-33 \pm 9$  W m<sup>-2</sup>, sensible heat flux of  $-20 \pm 9$  W m<sup>-2</sup>, longwave heat flux of  $-11 \pm 1$  W m<sup>-2</sup> and net heat flux of  $-3 \pm 17$  W m<sup>-2</sup>. The net solar radiation decreased shortly during the abrupt meteorological shift from 49 to 39 W m<sup>-2</sup> but continued to increase afterwards. The mean shortwave heat flux was  $44 \pm 5$  W m<sup>-2</sup> before the abrupt change in meteorological variables and  $58 \pm 6$  W m<sup>-2</sup> after the change.

Figure 1: Time series of (a) sea surface temperature, (b) sea surface salinity, (c) sea surface density, (d) temperature and salinity anomaly, (e) air temperature and wind speed, (f) heat flux components, (g) precipitation (P) and evaporation (E) rates on 14 March 2023.

## 3.1.2 Current velocities on the surface



To show the zonal (u component of velocity) and meridional (v component of velocity) velocity components and backscatter intensity from 07:30 to 09:30 UTC, we present time-depth plots derived from ADCP data collected on 14 March 2023 (Fig. 2). The zonal velocity component (Fig. 2a) shows a westward flow from 07:30 to 08:25 UTC, followed by an abrupt shift to an eastward flow with a pronounced current shear at an approximately 3 m depth. The meridional velocity component (Fig. 2b) indicates a northward flow from the surface to a depth of 2 m and a southward flow below 3 m during the same period. After 08:25 UTC, the flow shifted, with the surface current turning southward and the current below 3 m turning northward.

The backscatter intensity shows variations in signal strength across depth and time (Fig. 2c). It shows a noticeable shift at 08:20 UTC, with an increase in intensity near the surface. The observed signal coincides with the precipitation event and the period of increasing wind, suggesting that enhanced bubble entrainment in the upper water layer may have occurred. A distinct relationship is evident between the uppermost sampled bins (45 cm and 65 cm below the water surface) and the observed wind speed. Below the surface, the backscatter remains relatively stable with less pronounced changes over time.

Figure 2: Current velocity data measured with an ADCP mounted at the centre of HALOBATES on 14 March 2023. (a) Zonal current velocity, (b) meridional current velocity and (c) backscatter signals are shown in three panels, and the colour scale shows the velocity magnitude for each current velocity component.

#### 3.2. Harbour event on 15 March 2023



## 3.2.1 Thermohaline features during abrupt meteorological changes

On 15 March 2023, the local weather in the harbour of Bremerhaven underwent another abrupt meteorological change, with several abrupt shifts in air temperature, wind speed, and precipitation (Fig. 3e, g). During the operation of the ASV, the wind speed sharply increased to 6.53, 10.49 and 9.21 m s<sup>-1</sup> at 07:58, 08:29 and 10:41 UTC, respectively (Fig. 3e). Afterwards, notable decreases in air temperature of -0.5, -1.4 and -2.4 °C were observed at 08:14, 08:56 and 10:53 UTC, respectively (Fig. 3e, Table 1). The wind direction shifted multiple times from North–West to South–West (Fig. S4). During the most pronounced decrease in air temperature, the wind speed and wind direction stabilised, blowing steadily from the West–North–215 West to North–West. Coincidentally, with each abrupt meteorological change, the precipitation rate increased rapidly (Fig. 3g). The highest precipitation rate occurred with the largest decrease in air temperature, with a maximum precipitation rate of 1.24 mm h<sup>-1</sup>. The accumulated evaporation was consistently higher than the accumulated precipitation, indicating a positive evaporation–precipitation flux (Fig. S5).

The surface temperature showed three abrupt decreases concurrently with the observed reduction in air temperature and net heat flux and an increase in wind speed (Fig. 3a, e, and f). The temperature at the 100 cm depth was warmer over the entire observation than the skin layer temperatures, resulting in a cool skin layer of  $4.85 \pm 0.14$  °C. A higher temperature occurred at the 100 cm depth, with a mean temperature of  $5.22 \pm 0.14$  °C, resulting in a general skin anomaly temperature of  $-0.36 \pm 0.07$  °C. Following an increase in wind speed (Fig. 3e), the skin temperature anomalies reached their maximum at  $-0.36 \pm 0.07$  °C, at 08:35 UTC with a value of -0.63 °C, and at 10:56 UTC with a value of -0.54 °C.

A more saline skin layer was observed throughout most of the operational period compared with the 100 cm depth. The mean salinity during the observation was  $09.12 \pm 0.24$  g kg<sup>-1</sup> in the skin layer and  $9.07 \pm 0.07$  g kg<sup>-1</sup> at the 100 cm depth. A decrease in skin salinity was observed only at higher wind speeds, with values reaching those observed at the 100 cm depth, resulting in salinity anomalies close to zero (Fig. 3e). During the three abrupt meteorological changes with increasing wind speed (Fig. 3e), salinity anomalies were removed, and the skin layer and the 100 cm depth showed similar salinity. The density of the skin layer was consistently higher than that at the 100 cm depth, with a mean density of  $1,007.21 \pm 0.26$  kg m<sup>-3</sup> in the skin layer and  $1007.16 \pm 0.19$  kg m<sup>-3</sup> at the 100 cm depth. In general, the observed pattern aligned with the trend in salinity fluctuations.

Latent and sensible heat fluxes were consistent, with notable decreases occurring concurrently with periods of elevated wind speeds (Fig. 3f). The strongest reduction of the net heat flux of –252 W m<sup>-2</sup> was observed approximately 7 min later than the largest air temperature decreases and wind speed increases (10:41 UTC and 10:53 UTC). At that time, the net heat flux became negative (–79 W m<sup>-2</sup>) and the shortwave heat flux showed the largest decrease of –208 W m<sup>-2</sup>. The latent and sensible heat fluxes decreased notably from –13.36 and –4.93 W m<sup>-2</sup> to –72 and –50 W m<sup>-2</sup>, respectively.

Figure 3: Time series of (a) sea surface temperature, (b) sea surface salinity, (c) sea surface density, (d) temperature and salinity anomaly, (e) air temperature and wind speed, (f) heat flux components, (g) precipitation (P) and evaporation (E) rates on 15 March 2023.

## 3.2.2 Current velocities on the surface




From 08:00 to 11:45 UTC on 15 March 2023, the zonal velocity component indicated alternating flow patterns, with eastward and westward flows occurring intermittently throughout the observation period (Fig. 4a). Notable variability in the flow was observed over depth, with shear layers developing particularly at depths of approximately 3–4 m. The meridional velocity component exhibited a dynamic vertical structure (Fig. 4b). Early in the observation period, a northward flow dominated at greater depths. By contrast, a southward flow was more pronounced near the surface. As the period progressed, shifts in flow direction occurred, with the southward flow extending further into the water column and the northward flow retreating to lower depths. The backscatter intensity indicated clear temporal and depth-related variabilities (Fig. 4c). Enhanced backscatter values were evident near the surface after 09:30 UTC, suggesting that increased bubble generation and turbulence were likely increased by precipitation and wind-induced mixing. Similarly, a distinct association was observed between the uppermost backscatter bins and wind speed (Fig. S6). Below the 2.5 m depth, the backscatter intensity remained relatively stable.

Figure 4: Current velocity data measured with an ADCP mounted at the centre of HALOBATES on 15 March 2023. (a) Zonal current velocity, (b) meridional current velocity and (c) backscatter signals are shown in three panels, and the colour scale shows the velocity magnitude for each current velocity component.

## 3.3 North Sea event on 10 October 2022





## 3.3.1 Thermohaline features during abrupt meteorological changes

Thermohaline changes caused by precipitation for this event are described in Gassen et al. (2024b). Here, results are described concerning the changes in air temperature and associated heat exchange during the cold pool overpass at approximately 14:00 UTC (Fig. 5e–g). At this time, the air temperature decreased from 14.4 to 12.7 °C within 30 min (Fig. 5e). Before the abrupt meteorological change, the average air temperature was 14.51 ± 0.12 °C. After the abrupt change, the air temperature was 13.14 ± 0.05 °C (Fig. 5e). Between 14:20 and 14:30 UTC, the minimum wind speed was 5.8 m s<sup>-1</sup>, and the maximum wind speed was 15.0 m s<sup>-1</sup>, indicating a wind speed increase of 9.2 m s<sup>-1</sup> within 10 min, but delayed by approximately 15 minutes to the drop in air temperature. Before the overpass of the cold pool, the average wind speed was 8.79 ± 0.49 m s<sup>-1</sup>, increasing to 10.30 ± 1.04 m s<sup>-1</sup> afterward. At the same time, the wind speed increased, and the wind direction changed slightly from East to East–South–East (Fig. S7). After the abrupt meteorological change, the accumulated precipitation greatly exceeded the accumulated evaporation (Fig. S8).

General trends of the surface temperature and salinity in the skin layer and the 100 cm depth are discussed in Gassen et al. (2024b) and shown in Figure 5 for comparison with the other events (Fig. 1 and 3). The latent, sensible and net heat fluxes decreased by 48 W m<sup>-2</sup>, 32 W m<sup>-2</sup> and 100 W m<sup>-2</sup>, respectively. The shortwave heat flux was low over the entire time series due to overcast sky and increased at the end (between 15:00 and 15:20 UTC) to a mean value of 104 W m<sup>-2</sup> (Fig. 5f). At the end of the deployment, the net heat flux increased from -153 W m<sup>-2</sup> to -49 W m<sup>-2</sup>. The longwave heat flux was stable with slightly negative values, with a mean of  $-18 \pm 5$  W m<sup>-2</sup>. Approximately 18 minutes before the complete drop in air temperature and heat flux components, the peaks of temperature and salinity anomalies are reached. Notably, the salinity anomaly of about -0.15 g kg<sup>-1</sup> returns to near-zero values shortly after the precipitation ceases, whereas the temperature anomaly persists throughout the remaining observation period (Fig. 5d).

We conclude that the salinity and temperature anomalies are decoupled, with the temperature anomaly being influenced by the heat flux, while salinity anomalies are driven by precipitation, as discussed in Gassen et al. (2024b). At around 15:10 UTC, an increase in incoming shortwave radiation to 104 W m<sup>-2</sup> triggers an almost immediate response in the skin temperature, reducing its temperature anomaly. Although the increase in latent heat flux should cool the skin layer, this effect is overshadowed by the dominant warming from the increased shortwave radiation. Based on the findings of Gassen et al. (2024b) and the additional heat flux measurements in this study, we conclude that the temperature anomaly is primarily driven by the heat flux, likely due to the precipitation temperature is similar to that of the skin temperature. The modulation of the heat flux due to the cold pool was the driving force behind the transition from a warmer to a cooler skin layer.

Figure 5: Time series of (a) sea surface temperature, (b) sea surface salinity, (c) sea surface density, (d) temperature and salinity anomaly, (e) mean air temperature and wind speed with minimum and maximum values, (f) heat flux components and (g) precipitation (P) rates on 10 October 2022. Data from Gassen et al. (2024b) and figures modified and complemented with data on the net heat flux and its components.

#### 3.3.2 Current velocities on the surface

To highlight the zonal eastward (u) and meridional northward (v) velocity components and the backscatter intensity over the observation period from 13:00 to 15:15 UTC, from the surface to 5 m depth, we show time-depth plots derived from ADCP data collected on 10 October 2022 (Fig. 6). The entire profile of the maximum depth is shown in Figure S9. The zonal velocity component revealed a predominantly westward flow throughout the water column, represented by the negative values in blue. This westward flow intensified with depth, reaching magnitudes of approximately –0.4 m s<sup>-1</sup>. The meridional velocity component exhibited alternating flow directions over time and depth. Northward flows, depicted in red, and southward flows, shown in blue, occurred intermittently, with significant variability in the upper layers of the water column.

Conversely, in the deeper layers between 13:15 and 14:15 UTC, the currents predominantly shifted in a southward direction, indicating the presence of a significant shearing layer around a depth of 2 m. After 14:15 UTC, the currents shifted southward again, with notable shear observed in the first metre in the southward direction. The backscatter intensity plot reflects notable spatial and temporal differences in suspended particle concentrations (Fig. 6c). The backscatter intensity was elevated near the surface, denoted by the yellow and red regions (Fig. 6c), indicating higher particle concentrations or bubble generation by dynamic mixing processes, possibly driven by surface wind or wave activity. Below a depth of 2 m, the backscatter intensity was lower and more consistent, with minimal fluctuations over time. Overall, the plots provide a detailed depiction of the temporal and vertical variations in flow dynamics and particle distribution, offering valuable insights into the hydrodynamic processes occurring in the study area during the observed period.

Figure 6: Current velocity data measured with an ADCP mounted at the centre of HALOBATES on 10 October 2022. (a) Zonal current velocity, (b) meridional current velocity and (c) backscatter signals are shown in three panels, and the colour scale shows the velocity magnitude for each current velocity component. For a better comparison with the harbour events, only the upper 5 m are shown.

#### 3.4 Correlation analysis of temperature and salinity anomalies with meteorological parameters




Correlation analyses were calculated to determine the relationships between the temperature and salinity anomalies in the skin layer and the main atmospheric forcing, i.e., wind speed, air temperature, net heat flux, and precipitation intensity (Fig. 7). The correlation heat map after the Pearson correlation analysis from the harbour events on 14 and 15 March 2023 and from the North Sea event on 10 October 2022 shows that all events had a negative correlation between temperature anomalies and wind speed and a positive correlation between temperature anomalies and air temperature and net heat flux (Fig. 7a–c). Regarding these results, the correlation of the events on 14 March 2023 and 10 October 2022 was stronger than that on 15 March 2023. The temperature anomaly showed only a significant negative but weak correlation with precipitation on 15 March 2023 (Fig. 7b).

The correlation analysis of the harbour event on 14 March 2023 (Fig. 7a) revealed a weak correlation between salinity anomalies and meteorological variables. Wind speed demonstrated a significant, although weak, positive correlation with the salinity anomalies, while the net heat flux exhibited a weak and negative correlation with the salinity anomalies of the first harbour event. The North Sea measurements revealed significant positive correlations between salinity anomalies, air

temperature, and net heat flux. Conversely, a significant negative correlation was observed between salinity anomalies, wind speed, and precipitation intensity. The correlation analysis also showed a weak but significant positive correlation between salinity anomalies and backscatter signals during the harbour events. The North Sea event was the only event that showed a significant negative correlation between salinity anomalies and precipitation (Fig. 7c). As expected, the net heat flux showed a strong negative correlation with wind speed and air temperatures during all events (Fig. 7). The correlation analysis revealed positive but weak correlations between wind speed and backscatter signals, in line with the understanding that wind can influence suspended particles and turbulence in the water column.

Figure 7: Correlation heat map after the Pearson correlation analysis from (a) the harbour event on 14 March 2023, (b) the harbour events on 15 March 2023, and (c) the North Sea event on 10 October 2022. Each cell represents the correlation strength and direction between two variables, with different colours indicating the degree of correlation: warm colours (orange) for positive and cold colours (blue) for negative correlations. The colour intensity reflects the magnitude of the correlation, ranging from -1 (strong negative correlation) to +1 (strong positive correlation). Significant correlations (p < 0.05) are annotated with an asterisk.

## 345 4 Discussion







The findings of this study offer significant insights into the effects of abrupt meteorological changes on temperature and salinity anomalies and resulting heat fluxes in the presence of cold pools. All three events showed abrupt shifts in air temperature, wind speed, heat flux, and precipitation. The harbour events (14–15 March 2023) had a stronger decrease in air temperature than the North Sea event (10 October 2022). Changes in net heat flux, mainly driven by latent and sensible heat fluxes, coincided with air temperature and wind speed. These modulations in the heat exchange triggered by cold pools play a critical role in local energy budgets and can influence larger-scale climate patterns (Yu, 2007; Chou et al., 2000). Precipitation patterns varied across the events, affecting salinity only with the highest observed precipitation intensities. While the responses of the skin layer showed cooling, salinity responses indicated a complex interaction between wind speed and freshwater inputs. Temperature and salinity anomalies are further discussed in Sections 4.1. and 4.2. During harbour events, the skin layer density remained higher than in the NSL, indicating that the skin layer has special thermohaline properties. These density anomalies and their relationship with wind and evaporation are examined in Section 4.3.

#### 4.1 Effects on temperature anomalies

Across all three events, the sudden decreases in air temperature resulted in corresponding reductions in the sea surface temperature, with the skin layer exhibiting a more pronounced cooling than the 100 cm depth (Table 1). Resulting cool skin anomalies were of varying magnitudes and durations, and were forced by the drop in air temperature. This sudden drop in air temperatures is linked to the formation of atmospheric cold pools. These cold pools arise from the evaporation of precipitation, which cools the air mass (Zuidema et al., 2017). The dense air mass beneath the clouds descends and spreads across the surface in all directions. As they move outward, these cold pools are associated with the initiation of new atmospheric convection. In northern Germany, cold pools typically cause a temperature decrease of 3.3 °C (Kirsch et al., 2021), which falls within the range of our observations (0.5 °C to 4.3 °C). In addition to the primary driver of evaporative cooling of air masses beneath clouds, the downdraft of cooler and drier air masses from the mid-troposphere further enhances the evaporative cooling in these regional cold pools (Kirsch et al., 2021).

Except for the first abrupt meteorological change on 15 March 2023, which was less pronounced than the other events (Table 1), the decline in temperature within the skin layer exhibited a considerable decrease compared with the 100 cm depth in all other events. The average temperature change at the 100 cm depth corresponded to a disproportionately greater cooling by 48.4 % of the temperature change observed in the skin layer. This shows that the temperature change in the skin layer caused by abrupt meteorological changes is underestimated by almost 50 % when measured at a depth of 100 cm, which in turn leads to significant errors in the calculation of heat exchange between the ocean and atmosphere. Brilouet et al. (2023) present a numerical study to demonstrate that shallow convection systems, including cold pools, have a significant impact on the modulation of ocean layer processes.

In the context of fine-scale heat exchange processes, observations during the first harbour event on 14 March 2023 reveal interactions between atmospheric conditions and surface water temperatures. A positive net heat flux was driven by the solar radiation, despite the cloudy day. The skin temperature anomaly remained slightly negative due to precipitation cooling (Gosnell et al., 1995). This aligns with studies showing the skin layer's responsiveness to precipitation (Wurl et al., 2019; Gassen et al., 2024a; Gassen et al., 2024b). Despite an insignificant correlation between precipitation intensity and temperature anomalies, precipitation cooled the skin layer by 0.09 °C compared to 100 cm depth. The 4.5 °C cooling of air caused by the cold pool was notably strong compared to a long-term observation in North Germany (Kirsch et al., 2021) and significantly affected heat exchange dynamics. Sensible, latent, and longwave heat fluxes became negative, resulting in a net heat flux of – 3 ± 17 W m<sup>-2</sup>. Cloud shading affects heat exchange modulation (Brilouet et al., 2023) by reducing solar radiation.




Cloudy conditions during our observations suggest sensible and longwave heat fluxes mainly drove changes in heat exchange. For instance, sensible heat flux is significantly influenced by air temperature (Fairall et al., 1996b), meaning that greater temperature gradients between the air and the surface result in a higher sensible heat flux. In our study, the sensible heat flux varied by approximately 20-50 W m<sup>-2</sup>, compared to 15-20 W m<sup>-2</sup> during the passage of a tropical cold pool (Vickers and Mahrt, 2006) and the presence of a cumulus convection system (Yokoi et al., 2014). Conversely, the change in longwave heat flux with the passage of a cold pool was approximately 50 W m<sup>-2</sup> (e.g., Fig. 1), which is within the same range as convection 390 systems, i.e., 30-70 W m<sup>-2</sup> (Yokoi et al., 2024). Both sensible and longwave radiation are triggered by wind gusts associated with cold pools, which immediately cooled the skin layer, causing the temperature anomaly to decrease by 0.32 °C. Correlation analysis revealed positive correlations between temperature anomaly, air temperature, and net heat flux, while wind speed showed a negative correlation.

395 During the second harbor event on 15 March 2023, a cooler skin layer remained throughout the measurement period. A series of cold pools caused air temperature to drop, while wind speed increased simultaneously (Fig. 3). The largest decrease in air temperature was 2.4 °C, which was less than during the first event, although solar radiation was four times higher (mean  $258.18 \pm 80.53 \text{ W m}^{-2}$ ) than in the first event (mean  $60.14 \pm 12.50 \text{ W m}^{-2}$ ). A reduction in short-wave heat flux by at least 100 W m<sup>-2</sup>, due to cloud shading, cooled the skin layer, contrasting with the first event, where sensible and long-wave heat 400 fluxes, and likely precipitation, dominated the cooling effect due to the completely overcast sky and no reduction in incoming shortwave solar radiation during the cold pool's passage.

In northern Germany, typical midday incoming shortwave radiation of 1000 W m<sup>-2</sup> is reduced by approximately 62 % (to 380 W m<sup>-2</sup>) and by 83 % (to 170 W m<sup>-2</sup>) during cloudy and fully overcast skies (Liepert and Kukla, 1997), confirming the cloudiness during the first and second harbor events based on the different measured shortwave solar radiation (Fig. 1 and 2). This indicates that the driving forces cooling the ocean layer and consequently affecting heat exchange vary between cold pools, with cloud types and coverage being relevant alongside cooled air masses and wind gusts. Shading by clouds during the cold pool was most significant at the third cool pool observed during the second harbor event, triggering a reduction in net heat flux by 253 W m<sup>-2</sup> and consequently reversing the heat flux direction.

During the North Sea event on 10 October 2022, the skin temperature was, on average, slightly higher than the temperature at a depth of 100 cm, resulting in a positive temperature anomaly (i.e., a warmer skin layer) before the cold pool's passage (Gassen et al., 2024). Warmer skin layers typically develop under calmer sea conditions and high incoming solar radiation (Asher et al. 2014; Wurl et al. 2019), which were not present during this event. However, the warmer anomaly of approximately 0.03 °C was minor, and light rain prior our observation could have reduced the skin layer's density (Wurl et al. 2019) and absorbed heat from the air until near equilibrium (Fig. 5). As air temperature decreased and wind speed increased, the temperature anomaly reversed, leading to the formation of a cooler skin. This is supported by the strong positive correlation between temperature anomalies and net heat flux (Fig. 7c).

Wind speed showed a significant negative correlation with temperature anomalies. The North Sea event exhibited larger magnitudes in the correlation analysis compared to the harbor events, likely due to the higher wind speeds in the open sea environment. The observed cool skin temperature anomaly of approximately 0.15 °C for this event (Gassen et al., 2024) is similar to the observation reported by Donlon et al. (1999), which noted a cool skin with a mean temperature anomaly of 0.14 °C and wind speeds exceeding 6 m s<sup>-1</sup>, but without a cold pool. This suggests that the passage of cool pools does not necessarily further cool the skin layer but triggers an immediate reversal from a warm to cool skin layer with a change in net heat flux by 100 W m<sup>-2</sup>. Comparisons between the harbor and North Sea events should be made cautiously due to differing environmental settings and seasons.






The responses in thermal cooling of the skin layer and heat flux following cold pools are quasi-instantaneous, similar to the response of the skin layer to precipitation (Wurl et al., 2019). Interestingly, the cooling of the bulk temperature at one meter responded similarly fast during the stronger cold pools both in the harbour and the open North Sea. Bulk temperatures dropped by approximately 50% of the skin layer cooling. The rapid response of skin temperature in all events highlights the sensitivity of the ocean's uppermost layer to atmospheric changes (Wurl et al., 2019). Still, in this study, we conclude further that cold pools trigger the upward heat transport by instantaneous and dynamic mixing processes, as shown in the high backscatter of ADCP signals at the time of overpasses (Fig. 2, 4, and 6). This increased turbulence is vital for the vertical transport of heat, mass, and momentum within the ocean (Qiao et al., 2016).

Numerous studies have investigated the changes in sea surface temperature across various timescales, from daily to annual, primarily focusing on diurnal to seasonal variations (Folland and Parker, 1990; Cayan, 1992; Sura et al., 2006), but less on fine-scales ranging between seconds to minutes and at the skin layer with its immediate response. Gill and Niller (1973) concluded that temperature anomalies are predominantly driven by heat flux through the sea surface, and support the needs of future systematic studies on the skin layer. This study reveals fundamental processes occurring during the passage of cold pools. However, these first data are too limited to quantify and parameterize these processes due to the sporadic nature of cold pools, particularly in the mid-latitudes. In our case, we have conducted over 100 missions with HALOBATES in the North Sea and observed these three events.

Our findings underscore the critical importance of including the skin layer in future research, as changes induced by meteorological shifts could be underestimated when measuring depths of several metres, potentially leading to inaccurate analyses of surface temperatures and the effects of heat and gas fluxes (Börner et al., 2022). The significance of this research is further amplified by the potential increase in extreme weather events due to climate change (Allen and Ingram, 2002; Ummenhofer and Meehl, 2017). As Francis and Skific (2015) suggested, disproportionate Arctic warming could lead to more meridional characteristics in Northern Hemisphere circulation, potentially increasing the frequency of such events. Consequently, future studies should prioritize high-resolution measurements of the skin layer to capture these abrupt changes accurately. By including these fine-scale processes, a mechanistic understanding of heat fluxes in the skin layer can be enhanced.

#### 4.2. Effects on salinity anomalies







The harbour events showed variations in salinity in the skin layer and at the 100 cm depth passage of cold pools, although these changes were less pronounced compared to temperature fluctuations. The skin layer exhibited the highest change in salinity, except for the events on 14 March 2022 and the first event on 15 March 2022 (Table 1). The salinity in the harbour events does not exhibit a consistent trend, probably due to the harbour basin's geometry and the complex mixing process of salty North Sea water and freshwater from the river Weser. The basin can be regarded as a vast mesocosm, where both the wind fetch and the influx of mixed fresh and saline water from the river through the lock are limited.

By contrast, the North Sea event demonstrated a slight decrease in salinity, with the most pronounced reduction occurring in the skin layer (Gassen et al., 2024b). Although wind-induced mixing was stronger, the North Sea event demonstrated a decrease in salinity with the precipitation associated with the cold pool. The impact of precipitation on salinity anomalies a few centimeters below the surface (Asher et al., 2014; Boutin et al., 2014) and in the skin layer (Wurl et al., 2019; Gassen et al., 2023; Gassen et al., 2024b) is notable. In the absence of precipitation, the tropical skin layer is generally more saline by tenths of g kg<sup>-1</sup> due to the evaporative water loss (Wurl et al., 2019). When precipitation occurs, the skin layer's salinity changes most significantly as an initial response to the influx of freshwater compared to the bulk salinity, but with opposing evaporation processes.

The interaction between wind-induced mixing, precipitation intensity, and the freshening of the skin layer is complex (Wurl et al. 2019; Gassen et al. 2023, Gassen et al. 2024b). However, in mid-latitudes in the autumn, such as in the North Sea event, evaporation losses are limited, preventing the formation of a more saline skin layer. The finding underscores that temperature and its associated skin anomalies are more sensitive to a drop in air temperature and changes in heat flux during the passage of a cold pool. In contrast, salinity anomalies generally respond more to precipitation and return to their normal state quickly, especially in mid-latitude environments where evaporation processes may have a lesser impact on skin salinity.

## 4.3. Characteristics of the skin layer and their relation to density variations

The distinct thermohaline properties of the skin layer, which are known for their unique biogeochemical properties that differ from those of the underlying seawater (Wurl et al., 2011), have become particularly visible in these observations. The skin layer is a critical boundary between the ocean and the atmosphere, directly influenced by continuous exchange processes, such as heat, gas, and freshwater fluxes (Schlüssel et al., 1997). With increasing wind speed, evaporation rates also increase, leading to the removal of latent heat and freshwater and, consequently, the development of a cooler skin layer (Murray et al., 2000).

This cooling effect was notably observed during the harbour events and after the abrupt meteorological change in the North Sea event. During the harbour events, a cool and more saline skin layer was observed, which was more pronounced following abrupt meteorological changes. Cooler skin developed after abrupt meteorological changes during the North Sea event. Unlike in the harbour events, saltier skin was not observed because of the input of freshwater through precipitation.

In our observations, the skin layer was consistently denser than the 100 cm depth, with mean density differences of 0.05 kg m<sup>-3</sup> for the harbour events and 0.02 kg m<sup>-3</sup> for the North Sea event, underscoring its distinct characteristics. Similarly, Wurl et al. (2019a) observed a denser skin layer compared with the 100 cm depth in the tropical Pacific region. They found a density anomaly (skin – 100 cm) threshold of 13 kg m<sup>-3</sup>, at which interfacial tension could maintain the denser skin layer. Gassen et al. (2023) confirmed this, reporting density anomalies up to 10 kg m<sup>-3</sup>. Laboratory experiments and theoretical models further support these observations, demonstrating that denser fluids can float atop water surfaces due to interfacial tension (Singh and Joseph, 2005; Phan et al., 2012; Phan, 2014), reinforcing the unique interplay between the skin layer's physical properties and its response to environmental dynamics. Our observed density anomalies are much lower than those of Wurl et al. (2019a) and Gassen et al. (2023), suggesting that the surface tension is sufficient to maintain the denser skin layer over the NSL in our study, despite the increasing wind speeds and heat fluxes, which slightly increased the density differences.

## 495 **5. Conclusions**




The findings of this investigation offer significant insights into the effects of abrupt meteorological changes, i.e. cold pools, on temperature and salinity anomalies in both harbour and open sea environments at mid-latitudes. This study reveals the intricate relationships between atmospheric conditions and sea surface responses, providing valuable information for understanding air—sea interactions and their implications for local and regional climate dynamics. These observations highlight the complex and rapid response of the ocean skin layer to abrupt meteorological changes and show its distinct thermohaline properties, demonstrating its cooler and denser characteristics compared with underlying seawater, particularly in response to meteorological changes and varying environmental conditions. The temperature sensitivity of the skin layer to atmospheric forcing was almost twice as much as at a depth of 100 cm. This emphasises the importance of the skin layer in air—sea interactions.

Although we observed differences between the harbour and open sea environments, analysing the temperature anomalies enhanced the mechanistic understanding of small-scale processes at the air-sea interface. The harbour environment can be regarded as a large-scale mesocosm, providing a controlled yet dynamic setting for studying complex physical processes under natural conditions. This understanding is crucial for improving climate models and weather forecasting systems. In all cases, a significant and strong correlation was observed between the temperature anomalies and both air temperature and wind speed 510 at the air—sea boundary layer. A similar relationship was also identified in the heat fluxes at the boundary layer, underscoring the necessity of considering the combined influences of all factors to comprehensively understand the effects of meteorological events on surface conditions, thereby highlighting the dynamics of the skin layer. Our findings underscore the need for highresolution monitoring systems that capture abrupt changes in atmospheric and oceanic parameters, including the ocean's skin layer. This study emphasises the importance of further small-scale investigations of the skin layer during cold pool events in 515 other regions and of expanding the data in the mid-latitudes for a more robust comparison. Future studies could benefit from longer-term observations and more diverse geographical locations to better understand the variability in skin layer responses to these abrupt meteorological changes.

## Data availability

All the data generated or analysed during this study are available in this published article and on PANGAEA (Gassen et al., 520 2024c; Gassen et al., 2025).

#### **Author contribution**

LG took the lead in writing the manuscript, analysing data, and designing the figures. LG, OW, SMA, and JM conducted field observations and data recording. JM and OW contributed to the writing of the manuscript. JM contributed to the data processing and designing figures. SMA and LJ processed the data and contributed to the analysis of the results. OW supervised the project. All authors discussed the results and commented on the manuscript.

## **Competing interests**


The authors declare that they have no conflict of interest.

## Acknowledgements

We would like to thank the captain, the crew and the scientific team on board RV *Heincke* for their support, R. Henkel for carrying out the salinometer measurements, and the Alfred Wegener Institute for providing the salinometer.

## Financial support

The work on the RV *Heincke* was funded by the Ministry of Science and Culture of Lower Saxony as part of the project "The North Sea from Space: Using Explainable Artificial Intelligence to Improve Satellite Observations of Climate Change" (NorthSat-X), Project number VWZN3680. The analysis and the publication writing were carried out as part of the Freshwater Fluxes Over the Ocean (FreshOcean) project number 510639210.

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
