# Peer review of "Atmospheric cold pools abruptly reverse thermohaline features in the ocean skin layer"

_EGUsphere, 2025_

## Author Comment (AC1)

**Dew temperature on 14.03.2023 at several heights above the sea surface.**

[Figure]

**Dew temperature on 15.03.2023 at several heights above the sea surface.**

[Figure]

**Dew temperature on 10.10.2022 at several heights above the sea surface.**

[Figure]

**Time series of current velocity components u and v at 45 cm and 65 cm below the water surface, alongside wind speed, on March 14–15, 2023. Data are shown at 10-second resolution. The left y-axis represents current velocity (m s⁻¹), and the right y-axis represents wind speed (m s⁻¹).**

---

## Author Response (AR1)

**General changes by the authors:**

Based on the reviews and the request for a more in-depth discussion, the authors have decided to describe the abrupt meteorological changes as 'cold pools'. According to the literature and observations, the observed atmospheric events can be attributed to this phenomenon. This has been described in the revised text of the manuscript, including abstract, introduction, method, and discussion.

**Reviewer 1:**

This paper describes observations of the very surface of the ocean (skin layer as well as the subskin water), especially how temperature, salinity, current velocities and backscattering respond to abrupt changes in meteorology. The measurements were made from a state of the art, autonomous platform HALOBATES, and the intricate details observed are impressive.

However, I feel that as it stands, the paper is rather descriptive, and wish more scientific insights can be teased out from this novel dataset. For example, how good is the state of the art model for capturing the cool skin effect and changes in near surface hydrodynamics? Quantitatively to what extent and how quickly does the near surface hydrodynamics respond to atmospheric forcing? Was the atmosphere also responding to changes in the surface ocean (as might be expected for a coupled atmosphere-ocean system)? Or were you only seeing the atmosphere driving oceanic changes?

**Specific comments:**

| Comment                                            | Response                                                       |  |  |
|----------------------------------------------------|----------------------------------------------------------------|--|--|
| Line 32. Poor grammar: It's the skin layer         | Thank you very much for your comments.                         |  |  |
| that is response, not the characteristics that     | We have changed the text accordingly (l. 32).                  |  |  |
| are responsible.                                   |                                                                |  |  |
| Line 44. Whether precipitation enhances gas        | Indeed, this is an excellent point, and due to                 |  |  |
| transfer velocity (via mixing induced by           | the lack of observations, the effect of rain on                |  |  |
| falling droplets) or reduces gas transfer          | air-sea gas exchange is not well understood.                   |  |  |
| velocity (by making the surface ocean more         | But with a more careful literature review, we                  |  |  |
| stratified) depends on varies, depending on        | think that typically rain enhances the rate of                 |  |  |
| environmental conditions, I think.                 | CO 2 exchange, and can increase the global          |  |  |
|                                                    | oceanic CO 2 uptake (Asthon et al., 2016). We       |  |  |
|                                                    | have rephrased the text to "Precipitation                      |  |  |
|                                                    | typically increases the exchange of gases                      |  |  |
|                                                    | between the ocean and the atmosphere (Parc                     |  |  |
|                                                    | et al., 2024; Ho et al., 2004), and the global                 |  |  |
|                                                    | effect of this is an increase in the ocean's                   |  |  |
|                                                    | ability to absorb CO 2 of 6 % (Ashton et al.,       |  |  |
| Line 57 60 At a alamae this management             | 2016)." (1.43-45)                                              |  |  |
| Line 57-68. At a glance, this paragraph says       | We have rephrased the paragraphs to avoid                      |  |  |
| pretty much the same thing as the paragraph above. | repetitions (l. 52-76). We have included                       |  |  |
| auove.                                             | additional text to describe atmospheric cold pools (l. 54-65). |  |  |
| Line 90. Thickness of the skin layer sampled?      | The autonomous research vehicle                                |  |  |
| Is this a true skin temperature measurement?       | HALOBATES uses the rotating glass disk                         |  |  |
| 15 uns a true skin temperature measurement!        | TIALODATES uses the totaling glass disk                        |  |  |

| How can you be sure that the temperature of the sampled 'skin water' doesn't change during sampling (e.g. due to exposure to atmosphere during transfer)? Have you compared this temperature with typical measurements of skin temperature by IR methods?     | method to sample the skin layer (Wurl et al. 2024). The thickness of the skin layer sampled by the glass disks depends on the rotational rate (Shinki et al., 2012). A rotational rate of 10 rotations per minute is used for the glass disks collecting the skin layer with a thickness of approximately 78 ± 9 µm. The glass disks are covered with a semi-transparent acrylic glass hood to reduce changes in the measured temperature and salinity due to evaporative cooling. To prevent the heating of glass disks and the adherence of water, the hood features ventilation slits. A good agreement between concurrent SST measurements with an infrared radiometer and a glass disk assembly has been reported without the presence of a hood for low to moderate wind speed (Fig. 2b in Wurl et al. 2019). In addition, hoses are insulated with foam. |
|---------------------------------------------------------------------------------------------------------------------------------------------------------------------------------------------------------------------------------------------------------------|-----------------------------------------------------------------------------------------------------------------------------------------------------------------------------------------------------------------------------------------------------------------------------------------------------------------------------------------------------------------------------------------------------------------------------------------------------------------------------------------------------------------------------------------------------------------------------------------------------------------------------------------------------------------------------------------------------------------------------------------------------------------------------------------------------------------------------------------------------------------|
| Line 93. The order of the two numbers are reversed before respectively.  Line 107. I don't think Fairall et al 2003 is the correct reference for COARE3.6.  To compute heat flux in the COARE model, there is an option to 1) use subskin                     | We have changed the text accordingly (l. 101-102).  We have changed the references accordingly (l. 115-116).  In measuring skin temperature, we adopt the second option for calculating heat flux (see                                                                                                                                                                                                                                                                                                                                                                                                                                                                                                                                                                                                                                                          |
| temperature measurement and turn on the skin effect, or 2) use skin temperature measurement and turn off the skin effect. Which approach was used? Why not comparing the two approaches?                                                                      | also Jaeger et al., 2025). To maintain the focus of our study, we choose not to include a comparison here and instead refer to the extensive published work on the COARE algorithms. We appreciate the reviewer's thoughtful suggestion, and it could indeed be intriguing to compare both options with direct skin temperature measurements thoroughly. However, such a comparison extends beyond the scope of the study described in our manuscript.                                                                                                                                                                                                                                                                                                                                                                                                          |
| Line 107 how was shortwave measured?                                                                                                                                                                                                                          | The same method as the longwave, we will add it to the method section.  Net shortwave radiation was measured using an upward and a downward facing secondary standard pyranometer (SR20-D2, spectral range: 0.28 µm - 3 µm, Hukseflux, Netherlands) mounted on the vessel.                                                                                                                                                                                                                                                                                                                                                                                                                                                                                                                                                                                      |
| Figure 1d. one useful output from the COARE model is the cool skin effect, dter (or temperature anomaly here). It would be insightful to compare the modelled cool skin effect and the observation here. I'm a bit surprised that you have a cool skin at all | The phenomenon of a cooler skin is attributed solely to the heat flux in the skin layer. This is distinct from the net heat flux, with solar radiation adding the shortwave heat flux, which in turn warms up water at greater depths. This is the reason why Qnet can be                                                                                                                                                                                                                                                                                                                                                                                                                                                                                                                                                                                       |

during the first period of this event, given that all the heat fluxes seem positive (ocean heating). positive but Qskin can be negative. To calculate the net heat flux in the skin layer, it can be assumed that 15% of the solar radiation is absorbed within the first millimetre of the ocean, corresponding to the thickness of the thermal boundary layer (Hasse, 1971; Schmidt, 1908).

We have compared the thickness of the thermal boundary layer from the temperature anomaly and the heat flux in the skin layer, and reported a good agreement with the skin thickness computed by COARE3.6 (Jaeger et al., 2025).

Figure 1f, which component of the longwave heat flux is shown here? I'm somewhat surprised by its small magnitude as well as sign.

Figure 1f shows the net longwave radiation component of the surface heat flux budget, calculated as the difference between outgoing longwave radiation emitted by the surface and incoming longwave radiation from the atmosphere. The relatively small magnitude reflects the strong downward longwave flux associated with the overcast conditions on that day. Cloud cover enhances atmospheric emissivity, leading to increased incoming longwave radiation that can nearly offset surface emissions.

Here and for other examples, did you have any measurements of the atmospheric boundary layer structure? For this study, we have not obtained any measurements of the atmospheric boundary layer structure. We are aware that these measurements would help to analyse the response of the skin layer temperature to atmospheric forces and will include the measurements in future studies. For example, within our next expedition in the central Atlantic, we will take vertical profiles of air temperature, humidity, etc.

Line 185. Do you have further evidence that the increased backscatter is due to deposited particles? I would've guessed that increased wind speed led to more wave breaking and production of bubbles near the surface. What does a plot of 'surface' backscattering vs. wind speed look like? Does the depth of backscattering-'cline' increase with wind speed? Were there measurements of whitecap fraction?

We have analyzed the wind data in relation to the backscatter data from the surface layer bins (45 cm and 65 cm depth) and found a strong effect of wind speed on the backscatter signal. Accordingly, we revised the relevant sections of the manuscript (lines 239-242). We updated the conclusion to reflect that those bubbles, likely generated by increasing wind and possibly precipitation, affected the surface backscatter signal. We have also added this analysis to the supplementary material (see Fig. S6).

Because high-resolution wind data are not available for the offshore experiment, a more

Figure 2. The near surface ADCP measurements are very interesting. Perhaps more can be done with the data. For example how does current velocity and direction change with wind speed and direction? Were there temperature or density measurements over the first 5 m that gives an indication for the degree of stratification?

detailed analysis for this experiment could not be conducted.

We analyzed the upper two bins of current velocity in relation to the u and v components of wind speed. Our observations indicate that both wind direction and wind speed likely influence variations in current velocity within the upper water layer. However, due to the experimental setting within the harbor, boundary walls likely alter the flow patterns during the measurements, making it difficult to separate wind-driven effects from sitespecific influences reliably. For illustration, we provide a figure comparing the u and v components of wind speed with the respective current velocities (see attached figure). Given the substantial impact of harbor boundaries on current patterns, we did not include this figure in the manuscript to avoid potential misinterpretation of the data.

Section 3.4 I don't find this section very useful. The equations for estimating heat flux and ocean/atmosphere variables such as temperatures and winds are pretty well known. So of course there will be correlations.

See Figure below.

Thank you for your feedback on Section 3.4. We agree that the general relationships between heat flux and ocean-atmosphere variables are well known. However, the intention behind this section was not only to reproduce these relationships, but also to contextualise them within the specific conditions of our study domain timeframe. Correlations between some variables are also less clear, such as correlations between temperature and salinity anomalies with atmospheric variables. By quantifying these relationships using observational data and computed products, aimed to emphasise the relative importance and variability of the factors influencing our case, which may differ from general assumptions or other studies. Nevertheless, we have revised the text to make the motivation behind this analysis clearer and to de-emphasize the generally expected correlation.

With regard to how in water variables such as temperature anomaly, salinity, and backscatter respond to meteorological variables, one easy and potentially useful analysis may be a lag correlation analysis. Do in water variables respond immediately or is there a short time lag?

We appreciate the reviewer's suggestion regarding a lag correlation analysis. We explored this approach for two days with sufficient temporal resolution (14.03. and 15.03.), as it was not feasible for 10.10. due to the 10-minute sampling intervals. The analysis did not reveal a consistent or "clear"

Line 358-360. This strikes me as unlikely, as the air temperature was > water temperature (suggesting warmer precipitation temperature), and also the skin salinity > subskin salinity.

pattern across the datasets. For 14.03., which exhibited the most pronounced abrupt change in meteorological conditions, we observed a time lag of approximately 1:20 minutes between air temperature and SST (skin), and about 20 seconds between wind speed and SST (skin). While this suggests that short lags may exist during abrupt changes, the results are not systematic across the dataset and therefore cannot be generalized. We agree that this is an interesting aspect and will comprehensive consider more correlation analysis in future studies with higher temporal resolution.

Thank you for this comment. Precipitation temperature is typically related to the dew temperature rather than air temperature (Bui et al., 2019). Before the abrupt shift in weather conditions that occurred on 14 March 2023, the dew point temperature was approximately 3 °C lower than the air temperature. This suggests that the lower rain temperature was most likely due to evaporation. Please find a figure with the dew temperature of all events below.

Before precipitation begins, the skin layer exhibits higher salinity compared to the 1 m depth due to evaporation. As illustrated in Figure 3, the skin layer becomes less saline and reaches an equilibrium with the deeper layer with rainfall. Observational studies (Gassen et al. 2024, 2025) have demonstrated that the skin layer is primarily influenced by precipitation, depending on intensity and the size of the droplets.

Bui, A., Johnson, F., & Wasko, C. (2019). The relationship of atmospheric air temperature and dew point temperature to extreme rainfall. *Environmental Research Letters*, 14(7), 074025.

Section 4.1. This section is very descriptive still and doesn't really read like a discussion, but more of an extension to 'results.' The COARE model is known to be decent at reproducing the cool skin effect. Why not comparing the model vs. observation here and highlight places where the model may be improved? I understanding that the cool skin

Thank you for the feedback. We have extensively rewritten the discussion with a focus on cold pools and heat flux measurements.

| is dynamic, but  | does the | model | can | capture |
|------------------|----------|-------|-----|---------|
| the dynamics ali | eady?    |       |     |         |

Line 465. There is an important distinction between the thermal skin (or boundary) layer and the mass skin layer. Because heat diffuses much faster than mass, the thermal skin layer is quite a bit thicker.

It is indeed the case that there are differences between the thermal skin and the mass skin layer. The depth of the skin layer that is sampled is approximately  $80~\mu m$ . This depth is found within the range of both layers. The sentence will be changed to achieve greater clarity.

Dew temperature on 14.03.2023 at several heights above the sea surface.

Dew temperature on 15.03.2023 at several heights above the sea surface.

Dew temperature on 10.10.2022 at several heights above the sea surface.

Time series of current velocity components u and v at 45 cm and 65 cm below the water surface, alongside wind speed, on March 14–15, 2023. Data are shown at 10-second resolution. The left y-axis represents current velocity (m s-1), and the right y-axis represents wind speed (m s-1).

**Reviewer 2:**

The manuscript presents quite valuable data which is not easy to collect and may enlighten some of processes that are occurring in the first metres of ocean. The relevance of the research is high, the methodology is properly described, introduction is written well, however - as noticed by the other reviewer - I find the manuscript is too descriptive.

| Comment                                                                                                                                                                                              | Response                                                                                                                                                                                                                                                                                                                                                                                                                                                      |
|------------------------------------------------------------------------------------------------------------------------------------------------------------------------------------------------------|---------------------------------------------------------------------------------------------------------------------------------------------------------------------------------------------------------------------------------------------------------------------------------------------------------------------------------------------------------------------------------------------------------------------------------------------------------------|
| The discussion is weak, largely describing the results - can your results be related to other similar studies? Can your results improve quantification of the processes at the sea surface, and how? | Thank you for your feedback. We agree and discuss in the revised manuscript our findings in the context of cold pools. We focus in the revised discussion on the heat flux, and the differences in the response of the skin temperature versus skin salinity. We have better integrated the literature into our revised discussion.                                                                                                                           |
|                                                                                                                                                                                                      | Cold pools are very sporadic in their occurrence, and we have observed them by chance. In over 100 missions with HALOBATES in the North Sea, we found the presence of cold pools only in these three events. The data are too limited for quantification and improve air-sea parameterizations, but our study reveals the underlying fundamental processes in the skin and upper ocean layer triggered by the abrupt changes with the overpass of cold pools. |
|                                                                                                                                                                                                      | More systematic studies are required, for example in the tropical with the assessment of vertical atmospheric structures to understand and eventually quantify the processes.                                                                                                                                                                                                                                                                                 |
| Can you results be implemented in air-sea parameterizations, i.e., to improve the models?                                                                                                            | Thank you for the comment. Please see above.                                                                                                                                                                                                                                                                                                                                                                                                                  |
| What are bottlenecks of your approach (e.g., any problems in methodology, too sparse data,) and what can be done (if anything) to minimize them?                                                     | As outlined above, the methodology on skin layer observation is not straightforward and requires special platforms, such as HALOBATES. The sporadic nature of cold pools and the requirement for floating or autonomous platforms are not well-aligned, presenting the challenge of being in the right place at the right time with the appropriate equipment.                                                                                                |
|                                                                                                                                                                                                      | With a dedicated team of oceanographers, meteorologists, and modelers, these challenges can be addressed in the future. For instance, a more systematic study in the                                                                                                                                                                                                                                                                                          |

|                                             | tropics could help balance the sporadic nature |
|---------------------------------------------|------------------------------------------------|
|                                             | of cold pools with their frequency.            |
| Further, please clarify how you computed    | Pearsons correlation was computed using R      |
| Pearson's correlations - what is the data   | (Version 4.4.2). The entire datasets were      |
| interval around the event which is used for | analysed with the harbour event having a       |
| computations?                               | time interval of 10 seconds (n=744 and         |
|                                             | n=1482) and the North Sea event an interval    |
|                                             | of 10 minutes (n=15).                          |